# Methods for improving the identification of acute stroke during ambulance calls: A scoping review

Areej Almutairi[1,2☯*], Fadila Wirawan[1,3☯], Adam Lloyd[4☯], Tom Moullaali[5☯], Gareth Clegg[1,4☯]

**1** Usher Institute, University of Edinburgh, Edinburgh, Scotland, United Kingdom, **2** Department of Emergency Medical Services, King Saud Bin Abdulaziz University for Health Sciences, Jeddah, Saudi Arabia, **3** Department of Public Health Nutrition, Universitas Indonesia, Jakarta, Indonesia, **4** Scottish Ambulance Service, Edinburgh, Scotland, United Kingdom, **5** Centre for Clinical Brain Sciences, University of Edinburgh, Scotland, United Kingdom

☯ These authors contributed equally to this work.
\* Areejalmet@hotmail.com

## Abstract

### Background

Accurately identifying strokes during ambulance calls remains challenging, leading to low diagnostic accuracy and delays in dispatching appropriate services. Limited evidence exists regarding methods for improving call handlers' stroke recognition. This scoping review explores methods for enhancing stroke identification during emergency calls in ambulance control centres (ACCs).

### Methods

We conducted a scoping review following the methodology of the Joanna Briggs Institute and adhered to PRISMA-ScR guidelines. A systematic search was performed across five databases: Embase, Medline, Scopus, Web of Science, and CINAHL, also grey literature sources, covering publications from January 1964 to July 2024. We included studies that examined methods to improve stroke identification during emergency calls in ACCs. To assess the effectiveness of these methods, eligible studies must evaluate at least one of the following outcomes: accuracy of stroke diagnosis, time to diagnosis, effectiveness of staff training, and acceptability of identification techniques. Two reviewers independently screened the studies, extracted the data, and conducted an inductive thematic analysis to identify common themes.

### Results

Of the 3,619 studies identified, seven met the inclusion criteria. Included studies focused on technology and algorithms (n = 3), training and educational programs (n = 2), and improved triage tools (n = 2) to enhance stroke identification during emergency calls to ACCs. Studies on technology and algorithms have reported increased

**Data availability statement:** All relevant data are within the paper and its Supporting information files.

**Funding:** This study did not receive any funding. However, as it is part of the first author's PhD project, we acknowledge support through a PhD studentship provided by the University of Edinburgh and King Saud bin Abdulaziz University for Health Sciences.

**Competing interests:** The authors have declared that no competing interests exist.

**Abbreviations:** ACC, Ambulance Control Centre; EMS, Emergency Medical Services; PPV, Positive Predictive Value.

stroke identification sensitivity and positive predictive value (PPV) when using new algorithms compared to standard protocols. Training programs have led to improved dispatcher sensitivity in stroke recognition. Improved triage tools also reduce time-to-diagnosis and facilitate quicker emergency responses.

## Conclusion

This review highlights several methods for improving stroke identification in ACCs. Despite improvements in PPV, sensitivity, and diagnosis time, the lack of generalised standards, single-centre studies, and various population characteristics hinder broader impact. Future research should prioritise well-designed studies with standardised benchmarks to determine effectiveness, enabling effective prehospital stroke identification strategies.

---

## Introduction

Stroke is the second leading cause of death and disability worldwide, accounting for approximately 12 million deaths annually. According to the recent data, the incidence rate of stroke increased by approximately 70% between 1990 and 2021. The incidence rates for men were reported as 159 (143–177) and for women, 144 (130–159) per 100,000 individual-years [1]. This growing burden highlights the importance of understanding the stroke mechanism and rapid responses.

Stroke occurs due to an interruption in the blood supply to a part of the brain (ischaemic stroke) or bleeding from a ruptured blood vessel (intracerebral haemorrhage), leading to brain injury, disability, and often death [2,3]. Timely identification and access to emergency stroke care are essential to initiating a sequence of critical steps enabling rapid management leading to improved patient outcomes by reducing brain injury and subsequent medical complications. This 'stroke chain of survival' (See Fig 1), begins in the prehospital setting, where call handlers at the ambulance control centre rapidly gather crucial diagnostic information to dispatch the appropriate ambulance unit, ideally within the one-minute timeframe recommended by the American Heart Association [4].

However, accurately identifying stroke during ambulance calls is challenging because of the limitations of phone-based communication, which lacks the visual and physical assessments that are critical for recognising stroke symptoms. These challenges are exacerbated when patients have atypical stroke presentations or call handlers are inexperienced or insufficiently trained [5]. Stroke calls are often second-party calls in which laypersons or family members communicate with the ambulance service line on behalf of the patient, introducing additional communication complexities. These factors contribute to the low sensitivity and positive predictive value (PPVs) reported globally for stroke detection in call handlers. For instance, in the United Kingdom (UK), the sensitivity was only 51.5%, and the PPV was 12.8% [6], whereas in EMS Copenhagen, Denmark, the sensitivity was 66% and the PPV was 30% [7]. These findings highlight the urgent need for targeted training and

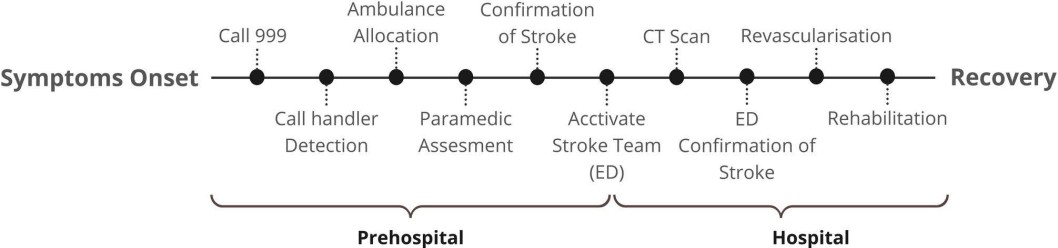

**Fig 1. Stroke chain of survival: critical steps required for optimal care, from emergency call to rehabilitation.** ED, emergency department. CT, computed tomography.

enhanced protocols for ambulance calls. These variabilities across different countries underscore the challenge of sub-optimal, accurate identification rates, even with a globally structured dispatch protocol.Current dispatch protocols, such as the Medical Priority Dispatch System (MPDS), which incorporates stroke screening tools such as the stroke diagnostic tool, continue to yield suboptimal identification rates. A retrospective study in Victoria, Australia, assessing the impact of stroke diagnostic tool introduction found no significant improvement in stroke identification during emergency calls, with rates remaining largely unchanged at 28.5% before and 28.2% after implementation [8]. These findings highlight the need for improved stroke recognition during emergency dispatches.

In response to these challenges, some studies, such as those by Richards et al. and Vuorinen et al., have analysed the phrases used to describe stroke during ambulance calls and identified linguistic patterns associated with missed stroke calls [9,10]. However, these studies primarily focused on analysing call data and identifying descriptive patterns rather than developing or evaluating strategies to improve stroke identification during calls. This leaves a critical gap in the research regarding actionable methods or interventions to optimise prehospital stroke identification during ambulance calls. To address this gap, we conducted a scoping review to systematically map existing methods for improving stroke identification in ACCs during ambulance calls. This review aimed to synthesise the evidence on stroke identification methods in ACCs, explore current approaches, characterise study types and outcomes, and identify gaps in the literature.

## Methods

### Review design

We chose a scoping review research design to explore existing evidence regarding stroke identification in ACCs comprehensively. This design is appropriate for meeting our study's aim because it has the flexibility to cover various study types and methods [11]. We adhered to the methodological framework outlined by the Joanna Briggs Institute (JBI) for Scoping Reviews [12].

### Protocol and registration

The Preferred Reporting Items for Systematic Reviews and Meta-analysis Protocol Extension for Scoping Review (PRISMA-ScR) was used to address our aim, methods, and reporting processes [13]. To maintain rigour and transparency, we registered our final protocol with the Open Science Framework [14].

### Information sources and search strategy

To identify relevant evidence, we developed a search strategy guided by the population, concept, and context (PCC) mnemonics. This framework helped to design our research question and structure the search terms. Using free text and MeSH terms, we searched the Embase, Medline, Scopus, Web of Science, and Cumulative Index to Nursing and Allied Health

Literature (via EBSCO) databases in July 2024. The search included English-language publications published from January 1964 to July 2024 (S1 Appendix). Although older studies might have reduced relevance to the current practice, we adopted an inclusive approach to capture the full breadth of research, given the limited evidence in the literature. Additionally, we searched grey literature sources that might provide evidence related to our scope, including the National Health Service (NHS), World Health Organization (WHO) Institutional Repository for Information Sharing (IRIS), American Heart Association (AHA), National Emergency Number Association (NENA), International Academies of Emergency Dispatch (IAED), and ProQuest Dissertations and Theses Global (PQDT Global).

### Selection criteria

We applied selection criteria to focus on studies that examined methods to enhance prehospital stroke identification in ACCs. The search included peer-reviewed journals and grey literature encompassing all study designs to ensure a thorough exploration of the topic. The inclusion of grey literature allowed us to capture current practices and innovations that might not yet be documented in academic publications. In addition, we conducted a scoping review because of the heterogeneity of the existing literature, which varies widely in terms of study design, population, intervention types, and outcomes.

As the review focused on studies of patients suspected of having had a stroke, the patients themselves or the person acting on their behalf must have called the ambulance telephone number and described symptoms resembling those of a stroke, or the call handler must have identified a potential stroke from their description.

We searched for studies that measured at least one of the following key outcomes: accuracy of stroke diagnosis, time to diagnosis, effectiveness of staff training, and acceptability of the identification techniques examined. These outcomes are crucial for evaluating the effectiveness of the methods used in the included studies, allowing for more comprehensive assessment of their impact and enabling the formation of well-supported conclusions regarding methods designed for optimising prehospital stroke identification during calls.

By adhering to these criteria, we aimed to identify studies that offer valuable insights into improving stroke identification during ambulance calls.

### Study allocation

We uploaded the results of the databases and grey literature searches to the Covidence website, which automatically removed duplicates [15]. Two research members (AA and FW) screened the titles and abstracts of all identified studies and independently reviewed the full text of all potentially relevant studies. Any disagreements regarding the study selection were resolved through discussion. We also checked the reference lists of all the included studies for any new and potentially relevant screening studies.

### Data charting

Two authors independently charted the data from each study, including author(s), publication year, study design, sample size, setting, stroke identification improvement method, outcomes, and key findings in a customised Excel spreadsheet. This sheet was piloted on a sample of five papers (See Table 1). The authors reviewed the charted data collaboratively and refined it as required.

### Data items

Data items included outcome measures that assessed intervention effectiveness through metrics, such as stroke identification accuracy, staff training efficacy, and acceptance of the implemented methods.

We considered ACC settings and stroke calls to understand the overall context of the studies. In addition, we reviewed the implementation details of the improvement methods, including the training duration (if mentioned) and tool usage, to assess their practical applications.

Table 1. Characteristics of the Included Studies. EMS, emergency medical services.

| Themes | Technology and Algorithms | | | Training and Education Programs | | Enhanced Triage Tool | |
|---|---|---|---|---|---|---|---|
| Author(s) | *Krebes et al., (2012)* [19] | *Scholz et al., (2022)* [20] | *Wenstrup et al., (2023)* [21] | *Watkins et al., (2014)* [22] | *Mattila et al., (2019)* [23] | *Malekzadeh et al., (2015)* [24] | *Dami et al., (2017)* [25] |
| Study Design | Mixed-method study. | Descriptive retrospective quantitative single case study. | Retrospective study. | Mixed-methods study. | Prospective observational cohort study. | Quasi-experimental study. | Retrospective observational analysis. |
| Sample Size | 5774 patients. | 9049 patients. | 155,696 calls. | EMS calls and hospital admissions. | 820 patients. | 246 patients. | 27,719 calls. |
| Setting | Berlin, Germany. | Copenhagen, Denmark. | Copenhagen, Denmark. | United Kingdom. | Helsinki, Finland. | Mashhad, Iran. | Vaud, Switzerland. |
| Approach for Identifying Stroke | A new dispatch algorithm was designed and tested to increase stroke recognition during calls. | ASR software is integrated into EMS dispatch to improve stroke detection during calls. | A machine learning model designed to increase stroke recognition during calls. | The training program is designed to recognise stroke during calls. | Identifying targets for improvement in dispatcher training for stroke recognition. | Implementing CPSS to improve stroke identification accuracy during calls. | A revised CPSS improve stroke identification during calls. |
| Reported Outcomes | Increased dispatcher sensitivity (53.3%) and PPV (47.8%). | Increased Stroke detection rate (61.19%). | Increased sensitivity (63.0%) and PPV (24.9%). | Increased dispatcher accuracy (80%). | The sensitivity identified was 72.0%. Among 46 missed stroke cases, 46 (47.5%) failed to recognise FAST symptoms, and (26.1%) could not evaluate symptoms. | Increased triage accuracy (77.7%) with CPSS. | Increased dispatcher sensitivity (67.8%) and PPV (9.4%). |
| Key Findings | The new dispatcher algorithm showed improved sensitivity and PPV in stroke recognition. | ASR software integration significantly increased stroke detection and thrombolysis rates, indicating that speech recognition technology can help dispatchers identify stroke cases. | The machine learning model significantly improved stroke recognition compared to human dispatchers. | The training program significantly increased dispatcher accuracy in stroke identification. | Identifying and targeting specific training needs, particularly in recognising speech disturbances, will significantly improve dispatcher recognition of stroke. | Implementing CPSS improved the accuracy of stroke identification during calls. | The revised CPSS improved stroke identification. |

EMD, emergency medical dispatcher; PPV, positive predictive value. ASR: Automatic speech recognition. CPSS: Cincinnati Prehospital Stroke Scale. FAST, face, arm drift, speech disturbance, and onset time.

The scope was restricted to handler-level interventions, excluding the in-hospital and on-scene Scenarios. To ensure extensiveness, the searches included 'dispatcher' and 'call handler because both terms can be used in different regions to refer to the same role. We also used the term ACC to refer to settings responsible for managing ambulance calls, which are also commonly referred to as emergency operation centres, emergency communication centres, or other similar terms across different countries, which were included in the search.

### Critical appraisal of individual sources of evidence

Although optional in scoping reviews, we conducted a critical appraisal to assess the reliability of the findings and to guide interpretations. The quality of the included studies was systematically evaluated using the JBI Critical Appraisal Checklist [16]. Themes and criteria common to all study designs were identified from the JBI checklists and combined into a unified evaluation tool applicable to all the included studies (See Table 2). Each study aligned with the aim of this review.

To maintain objectivity and reliability, multiple authors evaluated the grey literature based on the Authority, Accuracy, Coverage, Objectivity, Date, and Significance (AACODS) Checklist [17].

## Synthesis of the results

Two authors (AA and FW) independently charted and verified the data to ensure accuracy. Using an inductive approach to thematic analysis, we identified common themes by organising the data into three thematic categories: training programs, triage tools, and enhanced technology tools (See Table 1). We summarised the findings with narrative descriptions and tables.

# Results

## Selection of sources of evidence

A total of 3619 studies were reviewed, of which 75 underwent a full-text review to determine eligibility. Ultimately, seven studies were included in this review. Fig 2 shows the PRISMA-ScR flow diagram, showing the number of screened, assessed, and included studies, and reasons for exclusion [18].

## PRISMA diagram

## Characteristics of the sources of evidence

## Critical Appraisal

## Synthesis of the results

The findings of the included studies are summarised in Table 1, along with the identified themes that emerged: technology and algorithms, training and educational programs, and improved triage tools.

Table 2. Critical appraisal of the included studies. This table shows the results of the critical appraisal for each included study, using a unified tool adapted from the Joanna Briggs Institute Critical Appraisal Checklists to allow for consistent assessment across diverse study designs. Each criterion is marked using a colour-coded system to represent the appraisal outcome.

| Study \ Checklist  Qs | Krebes et al. (2012) [19] | Watkins et al. (2014) [22] | Malekzadeh et al. (2015) [24] | Dami et al. (2017) [25] | Mattila et al. (2019) [23] | Scholz et al. (2022) [20] | Wenstrup et al. (2023) [21] |
|---|---|---|---|---|---|---|---|
| Were the stroke call inclusion criteria roughly representative of the stroke population? | ✓ | ✓ | ✓ | ✓ | ✓ | ✓ | ✓ |
| Was the method used to identify strokes during calls valid for capturing all stroke cases, including atypical stroke presentations? | ✓ | ✓ | ✓ | ✓ | ✓ | ✓ | ✓ |
| Were the stroke identification method reliable results across all calls? | ✓ | ✓ | ✓ | ✓ | ✓ | ✓ | ✓ |
| Can the stroke identification method be tested in different settings? | ? | ? | ? | ? | ? | ? | ? |
| Were the outcomes measured to reflect the method's effectiveness (Accuracy of stroke diagnosis, time to diagnosis, staff training efficacy, acceptance of methods)? | ✓ | ✓ | ✓ | ✓ | ✓ | ✓ | ✓ |
| Were potential confounding factors (e.g., patient demographics, call complexity) identified? | ? | ✓ | ? | ? | ✓ | ✓ | ✓ |
| Was the statistical analysis appropriate for evaluating the effectiveness of the stroke identification method? | ✓ | ✓ | ✓ | ✓ | ✓ | ✓ | ✓ |

Symbols indicate

✓ = Meets the criteria. ? = Partially meets the criteria. ✗ = Does not meet the criteria.

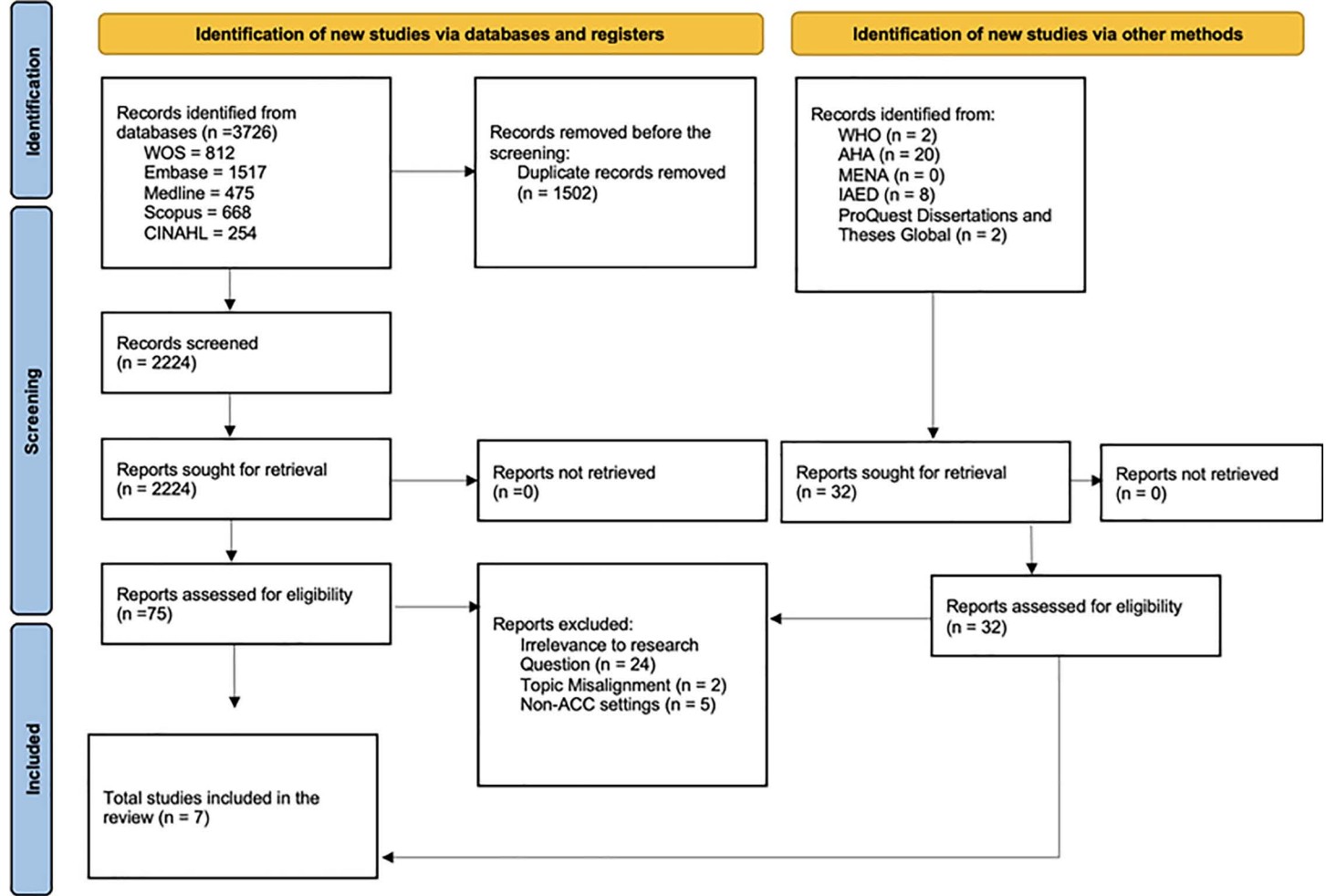

**Fig 2. PRISMA flow chart.** NHS, National Health Service. WHO, World Health Organisation, AHA, American Heart Association, MENA, National Emergency Number Association, IAED, International Academies of Emergency Dispatch. WOS, Web of Science. CINAHL, Cumulative Index to Nursing and Allied Health Literature.

**Technology and algorithms.** The first emerging theme is the use of new technologies and algorithms to improve stroke identification during ambulance calls. Wenstrup et al. examined automatic speech recognition software (ASR) with an integrated machine learning (ML) model to classify calls as related to stroke or non-stroke [21]. The ASR/ML model improved the sensitivity of stroke identification during emergency calls from 52.7% to 63.0% and the PPV from 17.1% to 24.9%. Similarly, Scholz et al. [20] investigated the potential of an ASR system previously used in an earlier study to identify cardiac arrests [26] and improve stroke detection. They estimated that implementing this system could result in a 61% increase in stroke detection rate and a 5% increase in thrombolytic administration. Krebes et al. developed and tested a stroke identification algorithm for a stroke emergency mobile unit based in Berlin [19]. They tested the algorithm's identification accuracy prospectively on patients admitted to the hospital by ambulance after first reviewing calls and found that it achieved a PPV of 47.8% (reported as an increase).

**Training and education programs.** The second theme that emerged was the training and educational programs. Watkins et al. developed an online training program for improving stroke identification accuracy based on an analysis of

stroke calls in UK call handlers and interviews with EMS providers [22]. The training program increased the sensitivity of stroke identification from the pre-training baseline by 63–80%. Mattila et al. conducted a study in Finland to examine the factors responsible for poor stroke identification during ambulance calls [23]. The key factors were the failure to generate high-priority dispatch codes, communication difficulties arising from poor phone reception, language barriers, and unclear speech. The presence of atypical stroke symptoms, such as collapse or falls, also decreases stroke identification. These factors were responsible for nearly half of all unidentified strokes during calls. These findings suggest that identification could be most effectively enhanced by better training call handlers in atypical stroke symptoms, clearer phone lines, and more extensive stroke dispatch algorithms that account for symptoms not included in the face, arm, speech and time (FAST) tool.

**Enhanced triage tools.** The final theme was the enhanced triage tools. Two studies focused on stroke tools, either by integrating an existing tool into a new system or enhancing it. The first such study was conducted by Dami et al. [25]. A modified Cincinnati Pre-hospital Stroke Scale (CPSS) was used to enhance dispatch-level stroke recognition by incorporating the time factor and concentrating on cases where symptoms began within the last five hours. The tool provides clear criteria for call handlers during calls, with specific exclusion criteria to rule out non-stroke cases (e.g., epilepsy) or cases with uncertain symptom onset. The tool produced a sensitivity of 67.8% but a very low PPV of 9.4%. Finally, Malekzadeh et al. compared the stroke identification accuracy of call handlers using the CPSS with the currently implemented National Guideline for Telephone Triage (NGTT) [24]. When assessed against subsequent hospital diagnoses, the CPSS yielded more accurate identification (77.7%) than the NGTT (65.6%).

**Grey literature.** The grey literature search yielded 35 records. After screening the titles and abstracts for relevance using the AACODS Checklist, we found that most studies were insufficiently focused on prehospital stroke identification within ACCs and were excluded. Two dissertations were reviewed in full text; however, they did not meet the review criteria [27,28]. As a result, no grey literature sources were included in this review.

## Discussion

### Summary of evidence

Our scoping review identified seven studies examining different methods for improving stroke identification during calls within ACCs. Three primary themes emerged: Technology and algorithms, training and educational programs, and enhanced triage tools. These findings offer valuable insights into current approaches to prehospital stroke identification and highlight several key areas for discussion.

### The significance of technology and algorithms

Technological tools, such as ML-enhanced ASR, can positively impact diagnostic accuracy. These audio-based approaches enable the leveraging of calls to classify potential strokes. Therefore, the identification rate increases. When such systems were used for calls, the PPV increased in Wenstrup et al. and by 16% in Scholz et al. by 8% [21,20]. Similarly, Krebs et al. reported a new dispatch algorithm with a sensitivity of 53.3% and PPV of 47.8% [19]. Although the performance figures are not yet optimal, they represent an improvement over traditional methods and highlight the potential of technology-driven approaches for enhancing stroke detection. However, enhancing the PPV is vital for early and accurate detection, reducing misdiagnosis, ensuring timely patient care, and optimising emergency resources. McClelland and Burrow noted that accurately identifying strokes can reduce on-scene time by approximately three minutes, which is crucial because of the rapid progression of stroke damage [6]. Audio-based technological tools, such as ML-enhanced ASR, can improve diagnostic accuracy, reduce on-scene time, and improve patient outcomes [29].

However, the figures reported by Wenstrup et al., Scholz et al., and Krebs et al. [20,19] cannot be confidently viewed as definitive improvements owing to the absence of standard benchmarks. However, external validation using real-world data may provide useful perceptions. For instance, the Scottish Stroke 2023 Annual Report reported a PPV of 18% and sensitivity

of 66% for ambulance stroke calls, indicating a diagnostic gap in which a number of stroke calls are misdiagnosed [30]. This decreased diagnostic accuracy underscores the need for optimisation methods to improve diagnostic reliability.

Without clear baselines or comparative standards, the observed increases in diagnostic accuracy are challenging to interpret as true progress. Additionally, these audio-only approaches rely solely on verbal cues, which may limit their effectiveness in thoroughly assessing stroke symptoms. Stroke-critical symptoms such as facial droop and limb weakness are often missed in audio-only ACC settings. Integrating visual-based methods, such as video triage, offers a promising extension to enhance stroke identification accuracy and improve decision making. Video triage systems enable detailed real-time visual assessments between paramedics and hospital-based neurologists or emergency physicians, thereby capturing symptoms which may not be identified in audio-only ACC settings [31,32].

Studies by Al Kasab et al., Bilotta et al., and Jacquin et al. demonstrated that video triage enables faster transport decisions and shorter prehospital times for patients requiring urgent care [33–35]. Similarly, Taqui et al. and Belt et al. found that combining video and standard telephone triage improved clinical assessment and decision-making by minimising delays associated with diagnostic uncertainty. They reported that the time to thrombolytic therapy was reduced to 23 min and 13 min, respectively [36,37].

It is possible to consider the video-triage method to not only compensate for the inherent challenges of audio-based methods with visual stroke symptoms but also to open the door for incorporating further multimodal methods, enhancing the accuracy and timeliness of stroke detection in prehospital care.

## Effectiveness of training and education programs

The second theme identified in our review was the effectiveness of structured, targeted training and education programs for enhancing call handlers' identification of stroke calls within ACCs. These programs have improved call handlers' competencies in three critical areas: recognising strokes, prioritising calls, and hastening ambulance dispatch. Watkins et al. showed that a training program increased stroke identification during calls by 20%, raising the identification rate from a baseline similar to that reported by Mattila et al. study, who found that inadequate training contributed to lower stroke recognition rates among call handlers [22,23]. In a related study by Sveikata et al., interactive EMS training improved PPV for stroke recognition by 8% in a population with a high cardiovascular risk [38]. These studies collectively underscore the importance of effective and targeted training programs to enhance call handlers' competencies, which are essential for timely stroke recognition and prompt allocation of emergency resources.

However, it is important to consider the diverse backgrounds of call handlers in various countries. In the UK (Watkins et al. settings [22]), call handlers are often non-clinicians, whereas in other countries, they may have a clinical background such as Nursing or Paramedicine. This variability could influence the effectiveness of training interventions and suggests that the implementation of focused training programs may need to be tailored based on the dispatcher's or handler's prior experience and expertise.

Although training alone has been shown to improve stroke identification, relying on a single intervention may not be sufficient to achieve significant and sustained improvements across diverse healthcare settings. The need for a combined approach is further supported by the fact that technological tools can augment the decision-making processes. For instance, ML-enhanced ASR systems can provide real-time support to call handlers by improving diagnostic accuracy. When paired with well-structured training programs, these tools can ensure that call handlers are better equipped to recognise and prioritise stroke calls quickly and accurately. Combining technology and training will likely result in a more robust and effective stroke identification process within the ACCs.

## Optimisation of triage tools

Refining triage tools to account for factors such as symptom onset and atypical stroke presentation can potentially improve stroke identification. Dami et al. showed that modifying the CPSS tool to prioritise symptom onset of less than

five hours could improve stroke identification accuracy, making the CPSS a more effective tool than the widely used FAST score (derived from CPSS) [25,39]. In the study by Berglund et al., FAST attained higher PPVs when used on-scene by the ambulance team than during calls. This suggests that in-person assessments allow for richer information gathering, implying that a more comprehensive tool, such as the modified CPSS, could provide more effective diagnostic guidance during calls [40].

However, the modified CPSS developed by Dami et al. had a sensitivity of only 67.8% and a very low PPV of 9.4% [25]. In stroke, where timely intervention is critical, higher sensitivity is preferred to minimise missed diagnoses. A low PPV indicates a high rate of false positives, potentially straining emergency resources. Although the tool incorporates the time factor and provides clear exclusion criteria, it may not capture the full complexity of stroke presentation. Therefore, future adaptations of the CPSS and similar tools must focus on enhancing sensitivity and specificity to achieve a better balance.

Furthermore, in Malekzadeh et al., the unmodified CPSS still offered better stroke identification accuracy than the National Guideline for Telephone Triage (NGTT), with (77.7%) of patients in the intervention group being correctly identified compared to 65.6% in the control group [24]. Although the difference between the two percentages is not significantly higher and the percentage is not optimal, it emphasises an area for potential improvement. However, it is important to note that the performance of these tools might be hindered or affected by the characteristics of the patient population and call handler training, as demonstrated by Baser et al. [41].

Therefore, while enhancing triage tools is critical, their success depends heavily on the context in which they are used. Dami et al. demonstrated that the modified CPSS showed moderate sensitivity but a very low PPV [25]. This suboptimal performance may be partly because call handlers, although nurses and paramedics with medical backgrounds, were not given specific training on the use of the modified tool before its implementation. Despite their clinical expertise, the lack of training on modifications may limit their ability to use the tool effectively. In contrast, Malekzadeh et al. attributed the higher accuracy of the CPSS to nurses who, besides having medical knowledge, received targeted training using the tool [24]. This finding suggests that specific training enhances the utility of the triage tools. This combination of preexisting knowledge and targeted training likely contributed to the superior performance of the tool in this study.

These findings highlight that, while refining triage tools is critical, their success is closely tied to the training of call handlers. Effective use of refined tools requires personnel to be equipped with the necessary skills through targeted training programs. Future efforts should prioritise the development of context-specific tools and provide training to enhance stroke identification accuracy during emergency calls.

## Interpretation of the findings

The findings of this scoping review have potential implications for clinical practice and policymaking. Stroke identification during emergency calls remains a critical challenge, with reported sensitivities as low as 51.5% and PPVs as low as 12.8% in certain systems. Well-targeted and integrated methods for improving stroke identification in ACCs can produce more effective and efficient resource use with significant impacts on patient outcomes. Enhanced identification tools, new algorithms, ML-enhanced ASR, and targeted training programs have resulted in improvements in sensitivity, PPV, and diagnostic accuracy. However, no single method can solve all the stroke identification challenges. The observed improvements suggest that there may be opportunities to enhance current systems for stroke detection during calls. However, the effectiveness of these approaches may vary across different healthcare settings and populations, and unintended consequences cannot be ruled out. Our review does not provide conclusive evidence that the included methods will substantially improve prehospital stroke identification during calls. Instead, it points out strategies that warrant further exploration through well-designed studies with standardised benchmarks. Based on these findings, we believe future research should prioritise evaluating the integration of these methods, identifying setting-specific implementation barriers, and developing standardised performance benchmarks. Clinicians and policymakers should consider these findings as the basis for pilot programs or additional research.

## Unanswered questions and future research

Although our review thoroughly examined the literature on optimisation methods for stroke identification during ambulance calls, several questions remain unanswered. For instance, although some studies, such as Wenstrup et al., compared their figures to baseline percentages and provided insights into potential improvements, the lack of consistent benchmarks or clearly defined standards across the included studies remains a significant challenge. This variability complicates efforts to evaluate the true effectiveness of these methods, making it difficult to determine their broader applicability or impact on stroke identification. Additionally, the sensitivity and specificity of these strategies, particularly for less common stroke types, e.g., posterior strokeand stroke chameleons require further exploration. Variations in healthcare infrastructure, patient populations, system settings, and call handler or dispatcher training or experience can lead to different performance outcomes across various contexts. Future research should prioritise evaluating these methods under standardised conditions with clear benchmarks to ensure more reliable and generalisable findings and to better quantify their true improvements in stroke identification.

## Strengths and limitations

Our study had several limitations. First, while the scoping review design allowed us to explore a wide range of study types and methods, the heterogeneity of the included studies, particularly in terms of population settings and methodological approaches, posed challenges in synthesising the findings into definitive conclusions. In addition, the small sample sizes and single-setting designs of some studies may compromise the generalisability of the results. Furthermore, the restriction to English-language publications may have introduced language bias, potentially excluding relevant research published in other languages.

Despite these limitations, our review was conducted using a thorough and exhaustive methodology. We developed a predefined protocol and implemented an extensive search strategy by using multiple databases. This approach enabled us to capture a broad range of published evidence and to systematically identify critical gaps in the existing literature. By meticulously synthesising findings from diverse studies, we constructed a robust analytical framework that provides a foundation for our conclusions. Our approach demonstrates a commitment to methodological transparency and scientific integrity, balancing a critical assessment of the available evidence with a systematic review process.

## Conclusion

This scoping review highlights several approaches to improve stroke identification during ambulance calls in ACCs, including ML-enhanced ASR, targeted training programs, and enhanced triage tools. Although these methods have shown improvements in PPV, sensitivity, and time-to-diagnosis, the lack of standardised benchmarks across these studies makes it difficult to determine the impact of these strategies. However, the effectiveness of these methods may vary based on local healthcare systems, patient demographics, and experience and training of dispatch personnel. Tailoring these methods to fit ACC settings is essential to optimise their impact and improve patient outcomes.

## Supporting information

**S1 Appendix. Summary of the search results yielded by the databases, including total records retrieved before and after deduplication.**
(DOCX)

**S1 File. PRISMA-ScR-Fillable-Checklist_10Sept2019.**
(DOCX)

## Acknowledgments

We thank the University of Edinburgh Library Team, Marshall Dozier, for helping us with the search strategy.

## Author contributions

**Conceptualization:** Areej Almutairi.

**Data curation:** Areej Almutairi.

**Formal analysis:** Areej Almutairi, Fadila Wirawan.

**Investigation:** Areej Almutairi.

**Methodology:** Areej Almutairi, Fadila Wirawan, Gareth Clegg.

**Project administration:** Areej Almutairi.

**Resources:** Areej Almutairi.

**Supervision:** Gareth Clegg.

**Validation:** Areej Almutairi.

**Visualization:** Areej Almutairi.

**Writing – original draft:** Areej Almutairi.

**Writing – review & editing:** Areej Almutairi, Adam Lloyd, Tom Moullaali, Gareth Clegg.

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
