## [Decision Letter · Decision Letter 0]

13 May 2025

Dear Dr. Almutairi,

Thank you for submitting your manuscript to PLOS ONE. After careful consideration, we feel that it has merit but does not fully meet PLOS ONE’s publication criteria as it currently stands. Therefore, we invite you to submit a revised version of the manuscript that addresses the points raised during the review process.

We look forward to receiving your revised manuscript.

Kind regards,

Nik Hisamuddin Nik Ab. Rahman

Academic Editor

PLOS ONE

Journal Requirements:

“This study was supported by the University of Edinburgh and the King Saud bin Abdulaziz University for Health Sciences. “

4. We note that Figure 1 in your submission contain copyrighted images. All PLOS content is published under the Creative Commons Attribution License (CC BY 4.0), which means that the manuscript, images, and Supporting Information files will be freely available online, and any third party is permitted to access, download, copy, distribute, and use these materials in any way, even commercially, with proper attribution. For more information, see our copyright guidelines: http://journals.plos.org/plosone/s/licenses-and-copyright.

5.  Please remove your figures from within your manuscript file, leaving only the individual TIFF/EPS image files, uploaded separately. These will be automatically included in the reviewers’ PDF.

Reviewers' comments:

Reviewer's Responses to Questions

**Comments to the Author**

1. Is the manuscript technically sound, and do the data support the conclusions?

Reviewer #1: Partly

Reviewer #2: Yes

2. Has the statistical analysis been performed appropriately and rigorously?

Reviewer #1: Yes

Reviewer #2: Yes

3. Have the authors made all data underlying the findings in their manuscript fully available?

Reviewer #1: Yes

Reviewer #2: Yes

4. Is the manuscript presented in an intelligible fashion and written in standard English?

Reviewer #1: Yes

Reviewer #2: Yes

Reviewer #1: The study is very vital and highlighted clearly the need for it but will be very essential to report about the gaps in the current practice or difficulties in the current practice with some data.

The other improvement, the aim was clear but the term objectives / objective were mentioned in at least three different places but not given importance to list them clearly in the report.

Reviewer #2: The authors present the results of a scoping review aimed at exploring methods to improve the identification of acute stroke during emergency calls at ambulance control centers (ACCs), explore current approaches, characterize study types and outcomes, and identify gaps in the literature. The authors identified seven studies examining different methods to improve stroke identification during calls at ACCs. The authors found that improved triage tools reduce time-to-diagnosis and facilitate faster emergency responses, highlighting several methods, including ML-enhanced ASR, targeted training programs, and enhanced triage tools to improve prehospital identification of stroke in ACCs. The study is potentially interesting, but can improved if the following considerations are addressed:

1.Please check and delete the colour sentences in the text (ex: “scene interventions” -page 16, line 4-)

2.In the Introduction section, it is worth adding epidemiological data on acute stroke (see and comment on the study published in Rev Esp Cardiol 2007; 60; 573-580). In this study, the cumulative incidence of cerebrovascular disease per 100,000 population was 218 (95% CI, 214-221) in men and 127 (95% CI, 125-128) in women.

3.Open lines of research include the exploration of methods to improve the identification of stroke versus stroke mimics, stroke versus stroke chameleons, or in subgroups of hemorrhagic versus ischemic stroke etiology, all important and challenging subsets of acute stroke that deserve to be evaluated in further studies.

4.The opinion of the authors on other future lines of research on this topic should be added in the text

**Do you want your identity to be public for this peer review?** For information about this choice, including consent withdrawal, please see our Privacy Policy

Reviewer #1: **Yes: ** Ahmed Ibrahim Al Kharusi

Reviewer #2: **Yes: ** Adrià Arboix

---

## [Author Response · Author response to Decision Letter 1]

3 Jun 2025

Reviewer #1: The study is very vital and highlighted clearly the need for it but will be very essential to report about the gaps in the current practice or difficulties in the current practice with some data.

The other improvement, the aim was clear but the term objectives / objective were mentioned in at least three different places but not given importance to list them clearly in the report.

Authors: Thank you for your comments. We appreciate your recognition of the relevance and necessity of this study. Regarding your suggestion to elaborate on the gaps or difficulties in current practice, we acknowledge that the existing literature directly focusing on our topic is limited and largely captured within the included studies. To address this, we carefully reviewed the broader but related literature and integrated selected studies to contextualise and build our argument more effectively. We have also added a new reference (lines 29-36) to further support and emphasise the gap we aim to address. We also thank you for your comment on the use of the term “objectives.” Upon revisiting the manuscript, we recognised that “objectives” was used inconsistently and did not add clarity. We have revised the text to consistently use the term “aim”, which better fits the structure and intent of this manuscript.

Reviewer #2: The authors present the results of a scoping review aimed at exploring methods to improve the identification of acute stroke during emergency calls at ambulance control centers (ACCs), explore current approaches, characterize study types and outcomes, and identify gaps in the literature. The authors identified seven studies examining different methods to improve stroke identification during calls at ACCs. The authors found that improved triage tools reduce time-to-diagnosis and facilitate faster emergency responses, highlighting several methods, including ML-enhanced ASR, targeted training programs, and enhanced triage tools to improve prehospital identification of stroke in ACCs. The study is potentially interesting, but can improved if the following considerations are addressed:

1.Please check and delete the colour sentences in the text (ex: “scene interventions” -page 16, line 4-)

Authors: Thank you for pointing this error out. Because our team regularly review the manuscript, some of the tracked changes were not deleted. We have fixed it, and we decided also to change the term “interventions” to “scenario” because we find it best to suit the sentence's meaning.

2.In the Introduction section, it is worth adding epidemiological data on acute stroke (see and comment on the study published in Rev Esp Cardiol 2007; 60; 573-580). In this study, the cumulative incidence of cerebrovascular disease per 100,000 population was 218 (95% CI, 214-221) in men and 127 (95% CI, 125-128) in women.

Authors: Thank you for your comment. We agree that including epidemiological data would strengthen the Introduction section. In response, we revisited our original reference and found that it has been updated with a more recent publication that was released earlier this year. Therefore, we chose to use updated data, which included incidence rates stratified by sex and per 100,000 individuals. These additions have been incorporated into the manuscript text on lines 3–7. We have also updated the reference list (Reference 1).

3.Open lines of research include the exploration of methods to improve the identification of stroke versus stroke mimics, stroke versus stroke chameleons, or in subgroups of hemorrhagic versus ischemic stroke etiology, all important and challenging subsets of acute stroke that deserve to be evaluated in further studies.

Authors: Thank you for your comment. Our “Unanswered questions and future research” section addresses diagnostic challenges, such as posterior strokes and variability in performance outcomes across contexts. We appreciate your suggestion to further acknowledge specific stroke subsets, such as mimics and ischaemic vs. haemorrhagic types, which are indeed relevant to the complexity of prehospital stroke identification. Thus, we have revised this section to mention these subgroups and the importance of studying methods that can improve detection across these diagnostic challenges (line 355).

4.The opinion of the authors on other future lines of research on this topic should be added in the text

Authors: Thank you for your comment. We agree that the authors opinions can help guide future research directions. In the “Interpretation of the findings” section, we outline our perspective on the implications of the current approaches and highlight areas for further exploration. To clarify our opinion, we have slightly revised the final sentences to directly reflect our team’s recommendations for future research, including the evaluation of real-world integration, standardisation, and implementation challenges (lines 343-346).

---

## [Editor Report · Decision Letter 1]

19 Jun 2025

Methods for Improving the Identification of Acute Stroke During Ambulance Calls: A Scoping Review

PONE-D-25-18093R1

Dear Areej Almutairi,

We’re pleased to inform you that your manuscript has been judged scientifically suitable for publication and will be formally accepted for publication once it meets all outstanding technical requirements.

Kind regards,

Nik Hisamuddin Nik Ab. Rahman

Academic Editor

PLOS ONE
---

## [Editor Report · Acceptance letter]

PONE-D-25-18093R1

PLOS ONE

Dear Dr. Almutairi,

I'm pleased to inform you that your manuscript has been deemed suitable for publication in PLOS ONE. Congratulations! Your manuscript is now being handed over to our production team.

Kind regards,

on behalf of

Professor Dr Nik Hisamuddin Nik Ab. Rahman

Academic Editor

PLOS ONE